



# Technical Note: Equilibrium droplet size distributions in a turbulent cloud chamber with uniform supersaturation

Steven K. Krueger[1]

[1]University of Utah, Salt Lake City, USA

**Correspondence:** Steven K. Krueger (steve.krueger@utah.edu)

**Abstract.**

In a laboratory cloud chamber that is undergoing Rayleigh-Bénard convection, supersaturation is produced by isobaric mixing. When aerosols (cloud condensation nuclei) are injected into the chamber at a constant rate, and the rate of droplet activation is balanced by the rate of droplet loss, an equilibrium droplet size distribution (DSD) can be achieved. We derived

analytic equilibrium DSDs and PDFs of droplet radius and squared radius for conditions that could occur in such a turbulent cloud chamber when there is uniform supersaturation. The loss rate due to fall out that we used assumes that (1) the droplets are well-mixed by turbulence, (2) when a droplet becomes sufficiently close to the lower boundary, the droplet's terminal velocity determines its probability of fall out per unit time, and (3) a droplet's terminal velocity follows Stokes' Law (so it is proportional to its radius squared). Given the chamber height, the analytic PDF is determined by the mean supersaturation

alone. From the expression for the PDF of the radius, we obtained analytic expressions for the first five moments of the radius, including moments for truncated DSDs. We used statistics from a set of measured DSDs to check for consistency with the analytic PDF. We found consistency between the theoretical and measured moments, but only when the truncation radius of the measured DSDs was taken into account. This consistency allows us to infer the mean supersaturations that would produce the measured PDFs in the absence of supersaturation fluctuations. We found that accounting for the truncation radius of the

measured DSDs is particularly important when comparing the theoretical and measured relative dispersions of the droplet radius. We also included some additional quantities derived from the analytic DSD: droplet sedimentation flux, precipitation flux, and condensation rate.

## 1 Introduction

In a laboratory cloud chamber, such as the Π Chamber at Michigan Technological University (Chang et al. 2016), it is possible

to produce Rayleigh-Bénard convection by applying an unstable temperature gradient between the top and bottom water-saturated surfaces of the chamber. Supersaturation is produced by isobaric mixing within the turbulent flow. When aerosols (cloud condensation nuclei) are injected at a constant rate, an equilibrium state is achieved in which the rate of droplet activation is balanced by the rate of droplet loss. After a droplet is activated, it continues to grow by condensation until it falls out (i.e., contacts the bottom surface).



Although the resulting equilibrium droplet size distributions (DSDs) have been extensively measured in the Π chamber, and theoretical models proposed for some aspects of the DSDs (e.g., Chandrakar et al. 2016, Chandrakar et al. 2017, Chandrakar et al. 2018a,b, Saito et al. 2019), obtaining a complete quantitative theory for the equilibrium DSDs has been elusive. The reasons for this include the difficulty of accurately measuring supersaturation in a cloud chamber (e.g., Chandrakar et al. 2016), as well

as uncertainties in our knowledge of the physical processes that determine the DSD. In particular, we don't know the relative importance of mean supersaturation and supersaturation fluctuations, nor do we have a quantitative understanding of droplet fall out.

In this study, we will assume that droplets grow subject to a uniform mean supersaturation, and that droplets fall relative to the turbulent flow at their Stokes' fall speed (for example, they are not affected by turbophoresis or thermophoresis). In section

1, we derive the equations which govern the evolution of the droplet radius and squared radius distributions, including the loss rate due to sedimentation. In section 2, we show how the equilibrium radius distribution is realized by using a Monte Carlo method, and compare the results to those that are obtained analytically in later sections. In section 3, we derive the analytic equilibrium solutions for the distributions and PDFs of radius and of squared radius, and from these obtain expressions for the median and mode radii. In section 4, we derive the first five moments of the radius from the analytic equilibrium PDFs,

including moments for truncated DSDs (those with positive lower limits). In section 6, we use statistics from a set of measured DSDs to check for consistency with the analytic DSD. We also demonstrate the importance of taking into account a non-zero truncation radius when comparing theoretical moments to moments from a measured but truncated DSD. In section 7, we present some additional quantities derived from the analytic DSD: droplet sedimentation flux, mean and PDF of the droplet residence time, precipitation flux, and condensation rate. Finally, section 8 contains the conclusions.

## 20  2  Governing equations

Our initial goal is to develop and solve the equations that govern the equilibrium droplet radius distribution under conditions that might be found in the Π chamber. Specifically, we will assume that (1) droplets grow subject to a uniform mean supersaturation, and (2) droplets fall relative to the turbulent flow at their Stokes' fall speed (for example, they are not affected by turbophoresis or thermophoresis).

### 25  2.1  Distribution of $r$

I will follow the notation used in Rogers and Yau (1989). They derived the following equation (their Eq. (7.31)) which governs the evolution of the droplet radius distribution, $v(r,t)$, subject to condensation:

$$\frac{\partial v(r)}{\partial t} = -\frac{\partial}{\partial r}\left(v\frac{dr}{dt}\right). \tag{1}$$

Here $v(r)\,dr$ is the number of cloud droplets per unit mass of air with radii in the interval $(r, r+dr)$. The condensational

growth rate is $dr/dt = \xi/r$, where

$$\xi = \frac{S-1}{F_k + F_d},$$





$S = e/e_s(T)$ is the saturation ratio, $e$ is the vapor pressure, $e_s(T)$ is the equilibrium vapor pressure over a plane water surface at temperature $T$, $F_k$ represents the thermodynamic term in the denominator that is associated with heat conduction, and $F_d$ is the term associated with vapor diffusion (Rogers and Yau 1989). The effects of droplet curvature and solute on droplet condensational growth are negligible because we consider only activated droplets (Rogers and Yau 1989, Siewert et al. 2017).

5    To generalize this to the cloud chamber in the presence of aerosol injection (which produces new droplets at a steady rate) and sedimentation (which removes droplets that fall to the bottom of the chamber), we add two terms to (1) so that it becomes

$$\frac{\partial v(r)}{\partial t} = -\frac{\partial}{\partial r}\left(\xi \frac{v}{r}\right) - v\frac{u}{h} + A(r), \tag{2}$$

where $u = k_1 r^2$ is the Stoke's Law droplet terminal velocity, $h$ is the height of the chamber, and $A(r)$ is the rate of production of (activated) droplets from the injected aerosol.

## 2.2   Distribution of $r^2$

Analogous to (1), the following equation governs the evolution of the squared radius distribution, $w(s,t)$, subject to condensation:

$$\frac{\partial w(s)}{\partial t} = -\frac{\partial}{\partial s}\left(w\frac{ds}{dt}\right). \tag{3}$$

Here $w(s)\,ds$ is the number of cloud droplets per unit mass of air with $s \equiv r^2$ in the interval $(s, s + ds)$. The condensational growth rate is $ds/dt = dr^2/dt = 2\xi$. When this is substituted into (3), the result is

$$\frac{\partial w(s)}{\partial t} = -2\xi\frac{\partial w}{\partial s}, \tag{4}$$

which has the form of the 1-D advection equation, with solution

$$w(s,t) = w_0(s - 2\xi t), \tag{5}$$

where the initial condition $w_0(s)$ is an arbitrary function. The solution (5) states that the initial distribution of $s = r^2$ simply translates at a rate $2\xi$ towards larger values of $r^2$ without any change of shape.

To generalize (4) to the cloud chamber in the presence of aerosol injection and sedimentation , we add two terms to (4) so that it becomes

$$\frac{\partial w(s)}{\partial t} = -2\xi\frac{\partial w}{\partial s} - w\frac{k_1}{h}s + B(s), \tag{6}$$

where $u = k_1 s$ is the Stokes' Law droplet terminal velocity and $B(s)$ is the rate of production of (activated) droplets from the injected aerosol.

## 2.3   Loss rate due to sedimentation

The probability that a droplet of radius $r$ will fall out due to sedimentation in a small time interval $\Delta t$ is $u/h\,\Delta t = k_1 r^2/h\,\Delta t$. This can be derived as follows: We assume that the droplets are well-mixed, in which case the $z$-coordinate of each droplet is a





random variable. Droplets are well-mixed if the turbulent flow velocities are predominantly larger than the terminal velocities of the droplets, in which case the droplets generally move with the flow. As a fluid element approaches the bottom wall, its vertical velocity approaches zero. However, a droplet in this fluid element will continue to fall at its terminal velocity. In a small time interval $\Delta t$, the droplet will fall a distance $\Delta z(r) = u\Delta t = k_1 r^2 \Delta t$. Therefore, all droplets with $z < \Delta z(r)$ will

reach the bottom ("fall out") during $\Delta t$. Because the droplets are well-mixed, a droplet's vertical coordinate $z$ may have any value between 0 and $h$. Therefore, a droplet's probability of falling out during $\Delta t$ is $\Delta z(r)/h = k_1 r^2/h\,\Delta t$, as stated above.

## 2.4    Related studies

Saito et al. (2019) derived governing equations for the distribution of $r^2$ in the presence of supersaturation fluctuations, both with and without mean supersaturation, and in which the droplet residence time is a specified constant for all droplets, rather

than depending on $r^2$ as in (6). Saito et al. (2019) also obtained analytical steady state PDFs of $r^2$ for these two governing equations.

     Garrett (2019) derived analytical steady-state size distributions of rain and snow particles from a governing equation similar to (2) in which the rain and snow particles grow from cloud droplets by collection and are lost by precipitation. However, collection differs from growth by condensation in that collection reduces the number of particles as the particles grow. To

represent both collection and precipitation realistically, Garrett included the dependence of fall speed on particle size.

## 3    Monte Carlo equilibrium solutions

The steady-state (equilibrium) radius distribution, $v(r)$, which is governed by (2), and the equilibrium squared radius distribution, $w(s)$, which is governed by (6), can each be obtained using a Monte Carlo method. Because we are interested in equilibrium solutions, the supersaturation will be steady and uniform, so that $\xi$ is a constant, and the aerosol injection rate

will be constant. Because $r^2$ increases at a constant rate due to condensation in this case, and because the fallout probability depends linearly on $r^2$, the relationship between the mathematical solution and the physical processes is more obvious for $r^2$ than for $r$, so we will apply a Monte Carlo method to determine the $r^2$ distribution, $w(r^2)$.

     A Monte Carlo method for solving (6) does so by calculating the injection, condensational growth, and fall out for many individual droplets as a function of time. We inject droplets with $r^2 = 1\ \mu m^2$ after equal time intervals. After injection, $r^2$ for

each droplet grows by condensation at a constant rate, $dr^2/dt = 2\xi$. As described previously in section 2.3, the probability that a droplet will fall out in a small time interval $\Delta t$ is $P = k_1 r^2/h\,\Delta t$. Fall out is implemented by removing a droplet after a time step if $P < X$, where $X$ is a uniformly distributed random number between 0 and 1.

     Figure 1 (left) displays the radius squared versus time for 150 droplets growing by condensation in 10% supersaturation. The frequency distribution of $r^2$ is easily obtained from the Monte Carlo results because it is equal to the average number of

droplets present in each $r^2$ interval at a given time. Figure 1 (right) compares the equilibrium frequency distributions of the radius squared from the Monte Carlo model (for 6000 droplets) and from the analytic solution (24) for the same parameters.





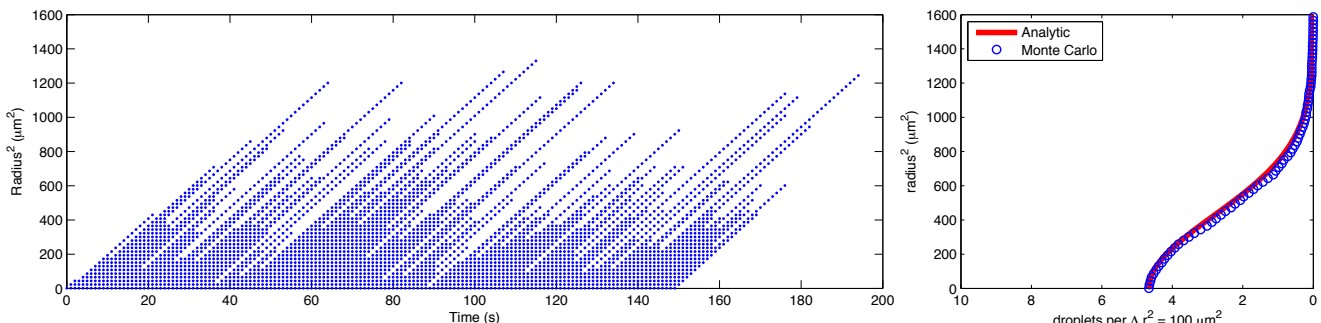

**Figure 1.** (Left) Radius squared versus time for 150 droplets growing by condensation in 10% supersaturation with probability of fallout per unit time of $u/h = k_1 r^2/h$ for $h = 1$ m. (Right) Frequency distributions of the radius squared from the Monte Carlo model (for 6000 droplets) and from the analytic solution (24) for the same parameters.

This confirms that (24) is indeed the equilibrium solution to (6). Note that the droplet injection interval (or rate) has no impact on the PDF of $r^2$.

The left panel of Figure 2 is the same as the left panel of Figure 1 except that the droplet fallout times are indicated by black circles. The droplet residence time, $\tau$, is the difference between the injection time, $t_i$, and the fall out time, $t_f$, and is practically

proportional to $r^2$ at the fall out time because

$$r^2(t_f) \approx r^2(t_f) - r^2(t_i) = 2\xi(t_f - t_i) = 2\xi\tau. \tag{7}$$

The frequency distribution of droplet residence times is easily visualized from the Monte Carlo results. Figure 2 (right) compares the frequency distributions of the droplet residence times from the Monte Carlo model (for 300,000 droplets) and from the analytic solution (58) for the same parameters. We used (7) to relate residence time to $r^2(t_f)$. Figure 2 (right) confirms that

(58) is the frequency distribution of the droplet residence times.

Figures 1 and 2 demonstrate that the $r^2$ and residence time distributions are closely related because each is determined by the stochastic nature of the droplet fallout process.

## 4 Analytic equilibrium solutions

We will now derive the analytic equilibrium solutions for the distributions of $r$ and $r^2$, $v(r)$ and $w(s)$, respectively.

### 4.1 Analytic equilibrium solution for the distribution of $r$

In a steady state, (2) becomes

$$0 = -\frac{d}{dr}\left(\xi\frac{v}{r}\right) - v\frac{k_1}{h}r^2 + A, \tag{8}$$





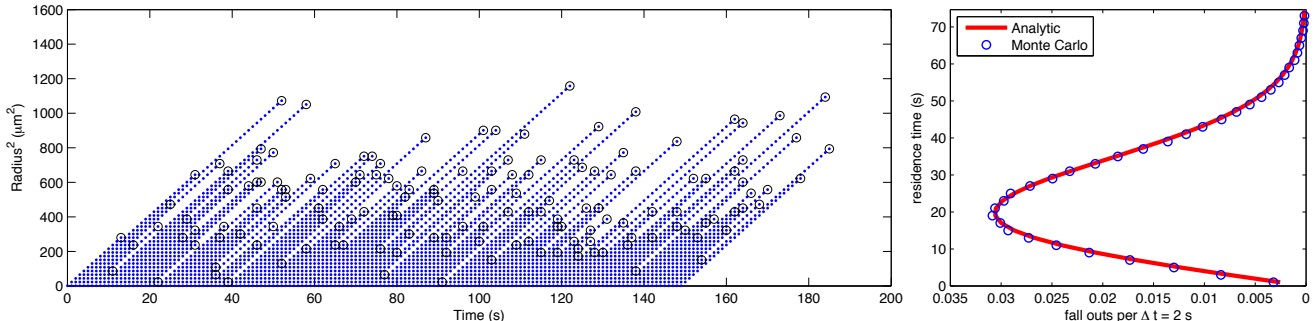

**Figure 2.** (Left) Same as the left panel of Figure 1 except that the droplet fallout times are indicated by black circles. (Right) Frequency distributions of the droplet residence time from the Monte Carlo model (for 300,000 droplets) and from the analytic solution (58) for the same parameters.

If the production of droplets by activation, $A(r)$, occurs only for $0 < r_0 < r < r_a$, and the loss due to sedimentation for $r < r_a$ is negligible, then we can integrate (8) from $r = r_0$ to $r = r_a$ to obtain

$$0 = -\int_{r_0}^{r_a} \frac{d}{dr}\left(\xi\frac{v}{r}\right) dr + \int_{r_0}^{r_a} A\, dr,$$

which becomes

5 $$0 = -\left(\xi\frac{v}{r}\right)\Big|_{r_0}^{r_a} + \int_{r_0}^{r_a} A\, dr,$$

then

$$0 = -\xi\left(\frac{v(r_a)}{r_a} - \frac{v(r_0)}{r_0}\right) + \int_{r_0}^{r_a} A\, dr,$$

and finally, using $v(r_0) = 0$,

$$\frac{v(r_a)}{r_a} = \frac{1}{\xi}\int_{r_0}^{r_a} A\, dr. \tag{9}$$

10 Eq. (9) allows us to consider the following o.d.e. instead of (8) for $r_a < r < \infty$:

$$0 = -\frac{d}{dr}\left(\xi\frac{v}{r}\right) - v\frac{k_1}{h}r^2, \tag{10}$$

with the boundary condition at $r = r_a$ given by (9). When the supersaturation is steady and uniform, $\xi$ is a constant so we can write (10) as

$$0 = -\frac{d}{dr}\left(\frac{v}{r}\right) - Cvr^2, \tag{11}$$





where $C \equiv k_1/(\xi h)$ is a constant with units of (length)$^{-4}$. The general solution to (11) is

$$v(r) = D\,r\exp(-C\,r^4/4),\tag{12}$$

where $D$ is an integration constant with units of (mass)$^{-1}$(length)$^{-2}$. Most of the solutions of the ordinary differential equations and integrals that appear in this study were obtained using Wolfram|Alpha (Wolfram Alpha LLC, 2019).

5 ## 4.2 Analytic equilibrium solution for the distribution of $r^2$

The derivation of $w(s) = w(r^2)$ is analogous to that for $v(r)$. In a steady state, (6) becomes

$$0 = -2\xi\frac{dw}{ds} - w\frac{k_1}{h}s + B(s),\tag{13}$$

If the production of droplets by activation, $B(s)$, occurs only for $0 < s_0 < s < s_a$, and the loss due to sedimentation for $s < s_a$ is negligible, then we can integrate (13) from $s = s_0$ to $s = s_a$ to obtain

$$0 = -\int_{s_0}^{s_a} -2\xi\frac{dw}{ds}\,ds + \int_{s_0}^{s_a} B\,ds,$$

which becomes

$$0 = -2\xi w\Big|_{s_0}^{s_a} + \int_{s_0}^{s_a} B\,ds,$$

then

$$0 = -2\xi\left(w(s_a) - w(s_0)\right) + \int_{s_0}^{s_a} B\,ds,$$

15 and finally, using $w(s_0) = 0$,

$$w(s_a) = \frac{1}{2\xi}\int_{s_0}^{s_a} B\,ds.\tag{14}$$

As before, we will consider the following o.d.e. instead of (13) for $s_a < s < \infty$:

$$0 = -2\xi\frac{dw}{ds} - w\frac{k_1}{h}s,\tag{15}$$

with the boundary condition at $s = s_a$ given by (14). When the supersaturation is steady, $\xi$ is a constant so we can write (15)
20 as

$$0 = -\frac{dw}{ds} - \frac{C}{2}ws.\tag{16}$$

The general solution to (16) is

$$w(s) = G\exp(-Cs^2/4)\tag{17}$$

where $G$ is an integration constant with units of (mass)$^{-1}$(length)$^{-2}$.





### 4.3 Droplet number concentration and integration constants

As already noted, $v(r)\,dr$ is the number of cloud droplets per unit mass of air with radii in the interval $(r, r+dr)$. Therefore, the number of cloud droplets per unit *volume* of air is

$$N = \rho \int_0^\infty v(r)\,dr = \rho D \int_0^\infty r\exp(-C\,r^4/4)\,dr = \rho D\frac{\sqrt{\pi}}{2\sqrt{C}} \tag{18}$$

where $\rho$ is the air density and $v(r)$ is given by (12). $N$ depends on both $D$ and $C$. We can solve (18) for the integration constant $D$ in (12):

$$D = N\frac{2\sqrt{C}}{\rho\sqrt{\pi}}. \tag{19}$$

To determine the integration constant $G$ in (17), integrate over the distribution $w(s)$ given by (17) to obtain

$$N = \rho \int_0^\infty w(s)\,ds = \rho G \int_0^\infty \exp(-C\,s^2/4)\,ds = \rho G\frac{\sqrt{\pi}}{\sqrt{C}}. \tag{20}$$

Eqs. (20) and (18) imply that $G = D/2$.

The number of cloud droplets per unit volume with radii larger than $a$ is

$$N(a) = \rho \int_a^\infty v(r)\,dr = \rho D \int_a^\infty r\exp(-C\,r^4/4)\,dr = \rho D\frac{\sqrt{\pi}}{2\sqrt{C}}\,\mathrm{erfc}(a^2\sqrt{C}/2) = N\,\mathrm{erfc}(a^2\sqrt{C}/2), \tag{21}$$

where $\mathrm{erfc}(z) \equiv 1 - \mathrm{erf}(z)$ is the complementary error function. From (21), we obtain the fraction of the total number of droplets with radii larger than $a$,

$$f(a) \equiv \frac{N(a)}{N} = \mathrm{erfc}\left(\frac{a^2\sqrt{C}}{2}\right). \tag{22}$$

### 4.4 PDFs of the equilibrium droplet size distribution

The PDF of the droplet radius distribution given by (12) is

$$p(r) = \frac{\rho\,v(r)}{N} = \frac{2\sqrt{C}}{\sqrt{\pi}}r\exp(-C\,r^4/4). \tag{23}$$

The PDF of the droplet squared radius distribution given by (17) is

$$q(s) = \frac{\rho\,w(s)}{N} = \frac{\sqrt{C}}{\sqrt{\pi}}\exp(-C\,s^2/4). \tag{24}$$

Both depend only on $C$. Figures 3 and Figure 4 display $p(r)$ and $q(s)$, respectively, for a supersaturation of 0.1% and $h = 1$ m.

By changing the independent variable from $s \equiv r^2$ to the non-dimensional variable $y \equiv s\sqrt{C}/2$, we obtain the non-dimensional PDF,

$$Q(y) = \frac{2}{\sqrt{\pi}}\exp(-y^2). \tag{25}$$




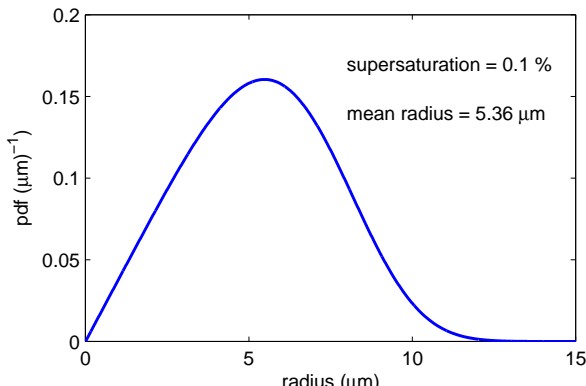

**Figure 3.** PDF of the droplet radius distribution given by (23) for a supersaturation of 0.1% and $h = 1$ m.

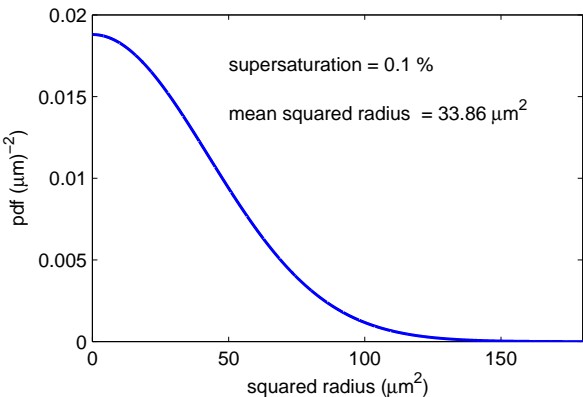

**Figure 4.** PDF of the droplet squared radius distribution given by (24) for a supersaturation of 0.1% and $h = 1$ m.

### 4.5 Median radius and CDF of the equilibrium droplet size distribution

The median radius, $\tilde{r}$, is defined by

$$\int_0^{\tilde{r}} p(r)\, dr = 0.5.$$

The cumulative density function (CDF) is the integral from 0 to $R$ of $p(r)$:

5    $$I(R) \equiv \int_0^R p(r)\, dr = \frac{2\sqrt{C}}{\sqrt{\pi}} \int_0^R r \exp(-C\, r^4/4)\, dr = \operatorname{erf}\left(\frac{\sqrt{C} R^2}{2}\right). \tag{26}$$

One can use (26) to determine $C$ given $R$ for any percentile $I$ of the cumulative distribution function. In general,

$$\sqrt{C} = \frac{2}{R^2} \operatorname{erf}^{-1}(I). \tag{27}$$





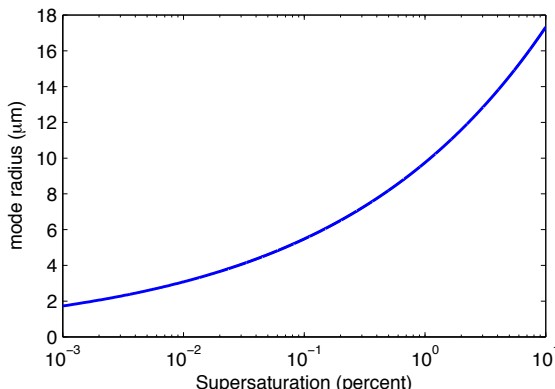

**Figure 5.** The mode radius versus the supersaturation for $h = 1$ m as given by (29).

If given the median radius, $\tilde{r}$, then $I = 0.5$ and

$$\sqrt{C} = \frac{2}{\tilde{r}^2}\mathrm{erf}^{-1}(0.5) \approx \frac{0.953873}{\tilde{r}^2}$$

so that

$$C \approx \frac{0.909873}{\tilde{r}^4}. \tag{28}$$

### 4.6 Mode radius

We derive the mode radius, $\hat{r}$, by expanding the derivative in (11) to obtain

$$0 = \frac{dv}{dr} - \frac{v}{r} + Cvr^3,$$

then applying $(dv/dr)_{r=\hat{r}} = 0$ and solving for $\hat{r}$:

$$\hat{r}^4 = \frac{1}{C} = \frac{\xi h}{k_1}. \tag{29}$$

The relationship between the supersaturation and the mode radius for $h = 1$ m is shown in Figure 5. This plot indicates that as the supersaturation increases by four orders of magnitude, from 0.001% to 10%, the mode radius increases from about 2 $\mu$m to 17 $\mu$m.

By writing (29) in the form

$$\frac{\hat{r}^2}{\xi} = \frac{h}{k_1\hat{r}^2}$$

we see that $\hat{r}$ is the droplet radius for which the timescale for droplet number growth due to condensation, $r^2/\xi$, equals the timescale for droplet number depletion due to sedimentation, $h/u = h/(k_1 r^2)$.





## 5 Moments derived from the analytic equilibrium PDFs

### 5.1 Mean radius

The mean radius is

$$\bar{r} = \int_0^\infty r\,p(r)\,dr = \frac{2\sqrt{C}}{\sqrt{\pi}}\int_0^\infty r^2\exp(-C\,r^4/4)\,dr = \frac{\sqrt{2}}{\sqrt{\pi}}\frac{\Gamma(\frac{3}{4})}{C^{1/4}}, \tag{30}$$

which depends only on $C$. Solve this for $C^{1/4}$ to obtain

$$C^{1/4} = \frac{\sqrt{2}}{\sqrt{\pi}}\frac{\Gamma(\frac{3}{4})}{\bar{r}},$$

so

$$C \approx \frac{0.913893}{\bar{r}^4}. \tag{31}$$

Eqs. (28) and (31) imply that

$$\frac{\tilde{r}}{\bar{r}} \approx 0.998898.$$

The mean radius of droplets with radii larger than $a$ is

$$\bar{r}(a) = \frac{\int_a^\infty r\,p(r)\,dr}{\int_a^\infty p(r)\,dr} = \frac{\frac{2\sqrt{C}}{\sqrt{\pi}}\int_a^\infty r^2\exp(-C\,r^4/4)\,dr}{f(a)} = \frac{\frac{\sqrt{2}}{\sqrt{\pi}}\frac{\Gamma(\frac{3}{4},\frac{a^4C}{4})}{C^{1/4}}}{\text{erfc}\left(\frac{a^2\sqrt{C}}{2}\right)} \tag{32}$$

where $f(a)$ is the fraction of the total number of droplets with radii larger than $a$ and $\Gamma(b,x)$ is the upper incomplete gamma function. Because

$$\int_a^\infty p(r)\,dr = \frac{N(a)}{N},$$

we can use

$$\frac{N(a)}{N} \equiv f(a) = \text{erfc}\left(\frac{a^2\sqrt{C}}{2}\right)$$

from (22). The upper incomplete gamma function is defined here as

$$\Gamma(b,x) \equiv \int_x^\infty t^{b-1}e^{-t}\,dt.$$

Note that the MATLAB® upper incomplete gamma function is defined differently, as

$$\frac{1}{\Gamma(b)}\int_x^\infty t^{b-1}e^{-t}\,dt,$$

and is called using `gammainc(x,b,'upper')`; note the reversed argument order.




## 5.2 Mean squared radius

The mean of the squared radius is

$$\overline{r^2} = \int_0^\infty s\,q(s)\,ds = \frac{\sqrt{C}}{\sqrt{\pi}} \int_0^\infty s\exp(-C\,s^2/4)\,ds = \frac{2}{\sqrt{\pi}\sqrt{C}} \tag{33}$$

which depends only on $C$. Solve for $C$:

$$C = \frac{4}{\pi(\overline{r^2})^2} \approx \frac{1.273240}{(\overline{r^2})^2}. \tag{34}$$

Eqs. (31) and (34) imply that

$$\frac{\overline{r^2}}{\bar{r}^2} \approx 1.180341. \tag{35}$$

The mean of the squared radius of droplets with radii larger than $a$ is

$$\overline{r^2}(a) = \frac{\int_{a^2}^\infty s\,q(s)\,ds}{\int_{a^2}^\infty q(s)\,ds} = \frac{\frac{\sqrt{C}}{\sqrt{\pi}}\int_{a^2}^\infty s\exp(-C\,s^2/4)\,ds}{f(a)} = \frac{\frac{2}{\sqrt{\pi}\sqrt{C}}\exp(-a^4 C/4)}{\mathrm{erfc}\left(\frac{a^2\sqrt{C}}{2}\right)}. \tag{36}$$

## 5.3 Mean cubed radius

The mean cubed radius is

$$\overline{r^3} = \int_0^\infty r^3\,p(r)\,dr = \frac{2\sqrt{C}}{\sqrt{\pi}} \int_0^\infty r^4 \exp(-C\,r^4/4)\,dr = \frac{2\sqrt{2}}{\sqrt{\pi}}\frac{\Gamma(\frac{5}{4})}{C^{3/4}} \tag{37}$$

which depends only on $C$. Solve (37) for $C$:

$$C = \frac{4}{\pi^{2/3}}\left(\frac{\Gamma(\frac{5}{4})}{\overline{r^3}}\right)^{4/3} \approx \frac{1.635767}{(\overline{r^3})^{4/3}}. \tag{38}$$

Eqs. (30) and (37) imply that

$$\frac{\overline{r^3}}{\bar{r}^3} = \frac{\pi\Gamma(\frac{5}{4})}{\Gamma(\frac{3}{4})^3} \approx 1.547460. \tag{39}$$

The mean cubed radius of droplets with radii larger than $a$ is

$$\overline{r^3}(a) = \frac{\int_a^\infty r^3\,p(r)\,dr}{\int_a^\infty p(r)\,dr} = \frac{\frac{2\sqrt{C}}{\sqrt{\pi}}\int_a^\infty r^4\exp(-C\,r^4/4)\,dr}{f(a)} = \frac{\frac{2\sqrt{2}}{\sqrt{\pi}}\frac{\Gamma(\frac{5}{4},\frac{a^4 C}{4})}{C^{3/4}}}{\mathrm{erfc}\left(\frac{a^2\sqrt{C}}{2}\right)}. \tag{40}$$

## 5.4 Mean $r^4$

The mean $r^4$ is

$$\overline{r^4} = \int_0^\infty r^4\,p(r)\,dr = \frac{2\sqrt{C}}{\sqrt{\pi}} \int_0^\infty r^5 \exp(-C\,r^4/4)\,dr = \frac{2}{C} \tag{41}$$





which depends only on $C$. Solve (41) for $C$:

$$C = \frac{2}{\overline{r^4}}. \tag{42}$$

Eqs. (30) and (41) imply that

$$\frac{\overline{r^4}}{\overline{r}^4} = \frac{\pi^2}{2\Gamma(\frac{3}{4})^4} \approx 2.188440. \tag{43}$$

The mean $r^4$ of droplets with radii larger than $a$ is

$$\overline{r^4}(a) = \frac{\int_a^\infty r^4 p(r)\,dr}{\int_a^\infty p(r)\,dr} = \frac{\frac{2\sqrt{C}}{\sqrt{\pi}}\int_a^\infty r^5 \exp(-C\,r^4/4)\,dr}{f(a)} = \frac{2}{C} + \frac{\frac{2a^2}{\sqrt{\pi}\sqrt{C}}\exp(-a^4 C/4)}{\operatorname{erfc}\left(\frac{a^2\sqrt{C}}{2}\right)}. \tag{44}$$

### 5.5   Mean $r^5$

The mean $r^5$ is

$$\overline{r^5} = \int_0^\infty r^5 p(r)\,dr = \frac{2\sqrt{C}}{\sqrt{\pi}}\int_0^\infty r^6 \exp(-C\,r^4/4)\,dr = \frac{4\sqrt{2}}{\sqrt{\pi}}\frac{\Gamma(7/4)}{C^{5/4}} \tag{45}$$

which depends only on $C$. Solve (45) for $C$:

$$C = \frac{4}{\pi^{2/5}}\left(\frac{\Gamma(\frac{7}{4})}{\overline{r^5}}\right)^{4/5} \approx \frac{2.365245}{(\overline{r^5})^{4/5}}. \tag{46}$$

The mean $r^5$ of droplets with radii larger than $a$ is

$$\overline{r^5}(a) = \frac{\int_a^\infty r^5 p(r)\,dr}{\int_a^\infty p(r)\,dr} = \frac{\frac{2\sqrt{C}}{\sqrt{\pi}}\int_a^\infty r^6 \exp(-C\,r^4/4)\,dr}{f(a)} = \frac{\frac{4\sqrt{2}}{\sqrt{\pi}}\Gamma(7/4,\frac{a^4 C}{4})}{\operatorname{erfc}\left(\frac{a^2\sqrt{C}}{2}\right)}. \tag{47}$$

## 6   Supersaturation inferred from measured moments

This study was motivated by the question of whether fluctuations in supersaturation are needed to explain the steady-state DSDs measured in the Michigan Tech turbulent cloud chamber (Π chamber) under conditions of constant aerosol injection rate. In this section, we use statistics from a set of measured DSDs to check for consistency with the analytic DSD, which was derived neglecting the effects of supersaturation fluctuations and deviations from Stokes' fall speed.

We will use statistics from a set of 11 DSDs with a wide range of droplet number concentrations (Chandrakar et al. 2018c)
that were measured by Chandrakar et al. (2018a) when the temperature difference between the the top and bottom boundaries was 19 K. The DSDs were measured using a phase Doppler interferometer and were truncated at a radius of 2.5 $\mu$m because smaller droplets were not reliably detected (Chandrakar et al. 2018b). Measurements were made over an interval of about 200 minutes for each DSD.



Do we expect droplet curvature and solute effects to significantly affect the measured DSDs? These effects should be negligible for droplets with radii greater than the critical radius, $r^*$. The size distribution of aerosol residuals (aerosols from evaporated droplets) for the measured DSDs is probably similar to that reported in Chandrakar et al. (2017)'s Figure 3, which indicates a mode diameter of 110 nm and a standard deviation of about 40 nm for the NaCl particles. According to Table 6.2 in Rogers

and Yau (1989), $r^* \approx 0.6 \, \mu$m for such aerosol particles. Because the truncation radius of 2.5 $\mu$m is much larger than $r^*$, droplet curvature and solute effects should be negligible.

Because the PDF of the equilibrium droplet radius distribution, (23), depends only on $C \equiv k_1/(\xi h)$, the moments of the PDF also depend only on $C$. The dependence of the first five moments on $C$ are given by (30), (33), (37), (41), and (45). Measurements of one or more moments would allow one to determine $C$.

However, measured DSDs are often truncated due to lack of detectability of small cloud droplets or difficulty in differentiating unactivated aerosol particles from small cloud droplets. To deal with such DSDs, we derived the dependence of the first five moments of the droplet radius on $C$ and the truncation radius, $a$. These are given by (32), (36), (40), (44), and (47). With these, one can determine $C$ from a moment and the DSD's truncation radius.

Knowing $C$, one can solve for the supersaturation, $S-1$, given $k_1$, $h$, and the thermodynamic parameter $(F_k + F_d)^{-1}$. If the

droplets fall at their Stokes' fall speeds, then $k_1$ is the Stokes' fall speed parameter, which is known. However, if the droplet fall speeds are affected by turbophoresis or thermophoresis, for example, then the fall speed parameter may not be known. Even if the actual fall speed parameter is unknown, it is still useful to calculate the supersaturation from $C$ using $k_1$ equal to the Stokes' fall speed parameter. We will call this the "nominal supersaturation."

In Figure 6 we plotted the mean radius, $[r]$, mean squared radius , $[r^2]$, and mean cubed radius, $[r^3]$, versus the nominal mean

supersaturation for $h = 1$ m for DSDs with no truncation (blue) and for DSDs truncated at $r = 2.5$ $\mu$m (red). The black dots indicate the nominal mean supersaturation values implied by $[r]$, $[r^2]$, and $[r^3]$ obtained from 11 measured DSDs truncated at $r = 2.5$ $\mu$m (Chandrakar et al. 2018c). The inferred nominal mean supersaturations range from 0.008% to 0.6%. The vertical green lines pass through the nominal mean supersaturation values implied by the measured $[r^2]$ values, and allow a visual assessment of the consistency of the supersaturation values implied by the three measured moments for each of the 11 DSDs.

If each DSD measured in the Π chamber was determined by the mean supersaturation alone, we would expect all three of the moments from a DSD to imply the same nominal mean supersaturation. However, even if moments of the analytic PDF derived in this study are consistent with the corresponding measured moments, that would not prove that supersaturation fluctuations were absent. It could be that the effects of supersaturation fluctuations on the PDF are nearly the same as those of the mean supersaturation, and are therefore difficult to discern. Or it could be that the effects are small despite the fluctuations being

significant due to a low correlation between the fluctuations of supersaturation and droplet radius (Chandrakar et al. 2016).

Figure 7 quantifies the degree of consistency of the three measured moments with the corresponding derived moments for truncation radii ranging from 0 to 3 $\mu$m. For each of the 11 DSDs, we used the supersaturation values implied by each of the three moments to calculate the mean and standard deviation of the implied supersaturation. We then calculated the average coefficient of variation of the implied supersaturation, which is plotted versus truncation radius in Figure 7. The

average coefficient of variation exhibits a pronounced minimum at $r \approx 2.3$ $\mu$m, which is nearly the same radius as the reported



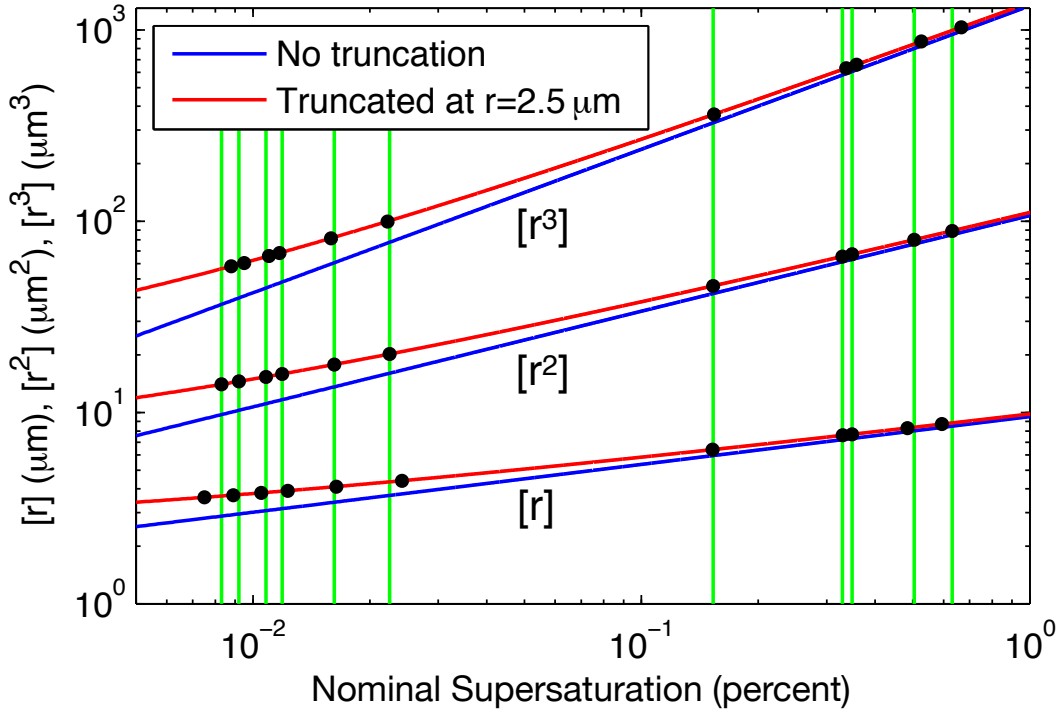

**Figure 6.** Mean radius, $[r]$, mean squared radius , $[r^2]$, and cubed mean radius , $[r^3]$, versus nominal mean supersaturation for $h = 1$ m for DSDs with no truncation (blue) and for DSDs truncated at $r = 2.5$ $\mu$m (red). The black dots indicate the nominal mean supersaturation values implied by $[r]$, $[r^2]$, and $[r^3]$ obtained from 11 measured DSDs truncated at $r = 2.5$ $\mu$m (Chandrakar et al. 2018c). The vertical green lines pass through the nominal mean supersaturation values implied by the measured $[r^2]$ values, and allow a visual assessment of the consistency of the supersaturation values implied by the three measured moments for each of the 11 DSDs.

truncation radius ($r = 2.5$ $\mu$m). Such agreement is expected if (1) the derived PDF is similar to the measured PDF and (2) the actual truncation radius is about 2.5 $\mu$m. The value of the average coefficient of variation at the truncation radius is a measure of the degree of consistency of the three measured moments with the corresponding derived moments. The value obtained ($\sim 2.5\%$) could be compared to values obtained using other PDFs, such as ones that include the effects of supersaturation fluctuations.

In Figure 7, the minimum value of the average coefficient of variation is less than 25% of the no-truncation value, which demonstrates that it is essential to consider the truncation radius when comparing theoretical moments to moments from a measured but truncated DSD. Figure 8 in section 6.2 adds further support to this conclusion.





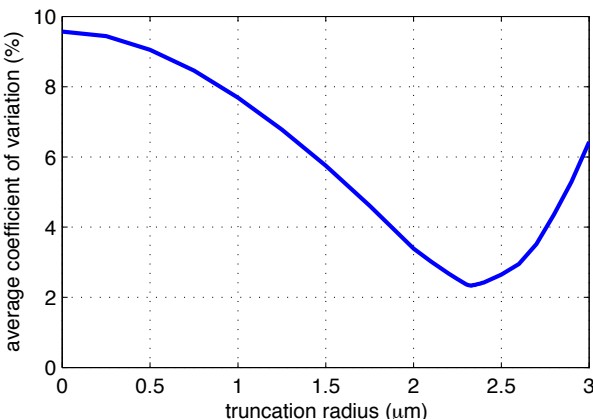

**Figure 7.** Average over the 11 DSDs of the coefficient of variation of the nominal mean supersaturation values implied by the three measured moments versus the truncation radius.

## 6.1 Standard deviation of the radius

The standard deviation is the square root of the variance. The variance of the radius is

$$\sigma^2 = \int_0^\infty (r - \bar{r})^2 \, p(r) \, dr = \frac{2}{\sqrt{\pi}\sqrt{C}}\left(1 - \frac{\Gamma(\frac{3}{4})^2}{\sqrt{\pi}}\right) \approx \frac{0.1724016}{\sqrt{C}} \tag{48}$$

which is easily obtained from the identity

$$\sigma^2 = \overline{r^2} - \bar{r}^2 \tag{49}$$

using (33) and (30). The variance of the radius for droplets with radii larger than $a$ is

$$\sigma(a)^2 = \bar{r}_s(a) - \bar{r}(a)^2 = \frac{2}{\sqrt{\pi}\sqrt{C}}\left(\frac{\exp(-a^4 C/4)}{f(a)} - \frac{\Gamma(\frac{3}{4}, \frac{a^4 C}{4})^2}{f(a)^2 \sqrt{\pi}}\right). \tag{50}$$

## 6.2 Relative dispersion of the radius

The relative dispersion of droplet radius, $\sigma/\bar{r}$, is obtained from (48) and (30):

$$\frac{\sigma}{\bar{r}} = \frac{\left[\frac{2}{\sqrt{\pi}\sqrt{C}}\left(1 - \frac{\Gamma(\frac{3}{4})^2}{\sqrt{\pi}}\right)\right]^{1/2}}{\frac{\sqrt{2}}{\sqrt{\pi}}\frac{\Gamma(\frac{3}{4})}{C^{1/4}}} = \frac{\pi^{1/4}\left(1 - \frac{\Gamma(\frac{3}{4})^2}{\sqrt{\pi}}\right)^{1/2}}{\Gamma(\frac{3}{4})} \approx 0.4246653. \tag{51}$$

The relative dispersion for a truncated DSD is obtained from (50) and (32).

Figure 8 displays the relative dispersion of the radius versus droplet number concentration, $n_d$. The measured values of dispersion are from DSDs truncated at $r = 2.5$ $\mu$m (Chandrakar et al. 2018a,b,c). The calculated values of dispersion used the average $C$ implied by the three measured moments for each of the 11 DSDs. They were obtained by assuming either DSDs





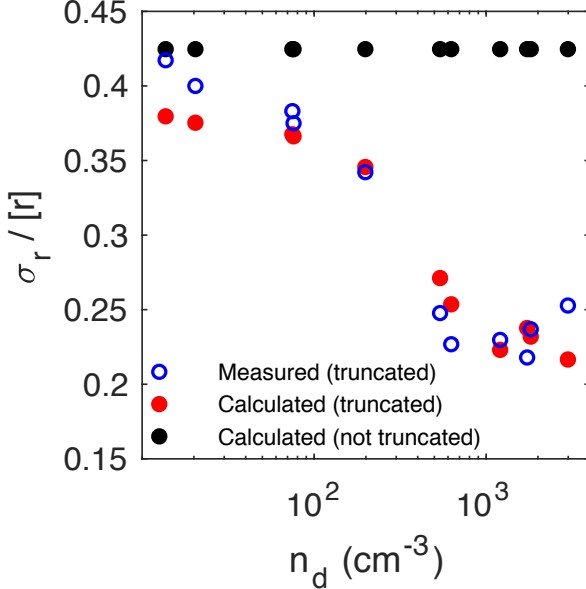

**Figure 8.** Relative dispersion of the radius versus droplet number concentration. The measured values of dispersion are from DSDs truncated at $r = 2.5\ \mu$m (blue circles) (Chandrakar et al. 2018c). The calculated values of dispersion used the average $C$ implied by the three measured moments for each of the 11 DSDs. They were obtained by assuming either DSDs truncated at $r = 2.5\ \mu$m (red dots) or not truncated (black dots) and used (50) and (32) or (51), respectively, with $h = 1$ m.

truncated at $r = 2.5\ \mu$m (red dots) or not truncated (black dots) and used (50) and (32) or (51), respectively, with $h = 1$ m. The calculated relative dispersion is constant ($\approx 0.425$) for no truncation, but is in good agreement with the measured values (which range from about 0.2 to about 0.4) when DSD truncation is accounted for. This is a dramatic example of the importance of considering the truncation radius when comparing theoretical moments to moments from a measured but truncated DSD. When confronted with these measurements of relative dispersion versus droplet number concentration, Chandrakar et al. (2018a,b) concluded that the results show that relative dispersion decreases monotonically with increasing droplet number density, and attempted to explain the results theoretically.

## 6.3 Standard deviation of the squared radius

The variance of the squared radius is

$$\sigma_s^2 = \int\limits_0^\infty (s - \overline{r^2})^2\, q(s)\, ds = \frac{2}{C}\left(1 - \frac{2}{\pi}\right) \approx \frac{0.7267605}{C} \tag{52}$$

which is obtained from the identity

$$\sigma_s^2 = \overline{r^4} - (\overline{r^2})^2 \tag{53}$$





using (41) and (33). The variance of the squared radius for droplets with radii larger than $a$, $\sigma_s^2(a)$, can be obtained from (53) using (44) and (36).

### 6.4 Relative dispersion of the squared radius

The relative dispersion of the squared radius, $\sigma_s/\overline{r^2}$, is obtained from (52) and (33):

$$\frac{\sigma_s}{\overline{r^2}} = \frac{[\frac{2}{C}\left(1-\frac{2}{\pi}\right)]^{1/2}}{\frac{2}{\sqrt{\pi}\sqrt{C}}} = \left[\frac{\pi}{2}\left(1-\frac{2}{\pi}\right)\right]^{1/2} \approx 0.7555106. \tag{54}$$

The relative dispersion of the squared radius for a truncated DSD can be obtained using (53), (44), and (36).

## 7 Some additional quantities

### 7.1 Droplet sedimentation flux

The droplet sedimentation flux, the number of droplets that exit the chamber due to sedimentation per unit area and time, is

$$F_{\text{sed}} = \rho D \int\limits_0^\infty u(r)\, r \exp(-C\, r^4/4)\, dr = N k_1 \overline{r^2}. \tag{55}$$

This result says that the droplet sedimentation flux is the same as if all droplets fell at the speed of one with the r.m.s. droplet radius.

### 7.2 Droplet residence time: mean and PDF

The mean droplet residence time, $\overline{\tau}$, is given by Eq. (1.45) in Nauman and Buffham (1983):

$$\overline{\tau} \equiv \frac{hN}{F}, \tag{56}$$

where $F$ is the total droplet flux, including the fluxes due to turbophoresis and thermophoresis. *We will assume that $F = F_{\text{sed}}$* so that

$$\overline{\tau} = \frac{hN}{F_{\text{sed}}} = \frac{h}{k_1\overline{r^2}} = \left(\frac{\pi h}{4k_1\xi}\right)^{1/2}. \tag{57}$$

This follows from using (55) for $F_{\text{sed}}$ and (33) for $\overline{r^2}$. The mean residence time in this case depends upon the chamber height, the Stokes' fall speed coefficient, and the supersaturation. Figure 9 shows $\overline{\tau}$ versus the supersaturation for $h = 1$ m. Figure 6 shows that the range of nominal mean supersaturations inferred from the measured moments is 0.008% to 0.8%. Figure 9 indicates that $\overline{\tau}$ decreases from about 900 s to 90 s over this range of actual supersaturations.

We noted in section 6 that the actual fall speed parameter, $k_1'$, is unknown. However, it could be determined from measurements of $\overline{\tau}$, $\overline{r^2}$, and $N$ by using (55) and (56):

$$k_1' = \frac{hN}{\overline{\tau}\,\overline{r^2}}. $$





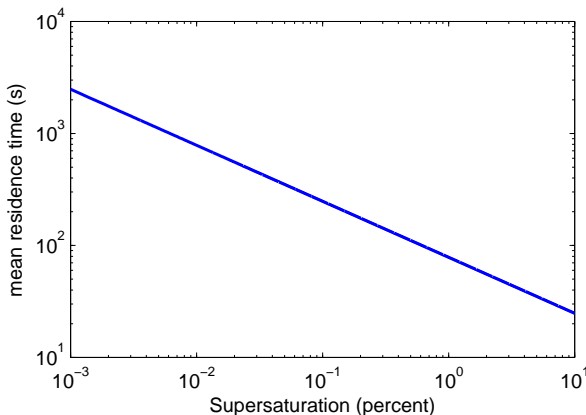

**Figure 9.** Mean droplet residence time versus supersaturation for $h = 1$ m as given by (56).

To derive the PDF of droplet residence times, $R(\tau)$, we start with the probability for a droplet of radius $r$ to fall out in a small time interval, $dt$, which we derived in section 2.3, then use $r^2 = 2\xi t$ to obtain

$$\frac{k_1 r^2}{h} dt = \frac{2k_1 \xi}{h} t \, dt \equiv b \, t \, dt.$$

This means that during a time interval $dt$, a fraction $b \, t \, dt$ of the droplets fall out. If $n(t)$ is the number of droplets injected at

$t = 0$ that remain at time $t$, then

$$\frac{dn}{dt} = -b \, t \, n,$$

which has the solution $n(t) = n(0) \exp(-b t^2 / 2)$. The distribution of droplet residence times is therefore $dn(t)/dt$, which we normalize to obtain the PDF of droplet residence times,

$$R(\tau) = b \tau \exp(-b \tau^2 / 2). \tag{58}$$

We verified that the mean droplet residence time obtained from (58) agrees with (56). Figure 10 displays the PDF of droplet residence times, $R(\tau)$, for 0.1% supersaturation and $h = 1$ m. In Figure 2 (in section 3), we compared $R(\tau)$ from a Monte Carlo method and $R(\tau)$ from (58) for 10% supersaturation.

### 7.3   Precipitation flux

The precipitation flux, the mass of liquid water that exits the chamber due to sedimentation per unit area and time, is

$$P = \rho \int_0^\infty u(r) \, m(r) \, v(r) \, dr = N \int_0^\infty u(r) \, m(r) \, p(r) \, dr.$$

Use $u(r) = k_1 \, r^2$ and $m(r) = \rho_L \, 4/3 \pi r^3$, the mass of a droplet of radius $r$, to obtain

$$P = N \, k_1 \rho_L \, \frac{4}{3} \pi \int_0^\infty r^5 \, p(r) \, dr = N \, k_1 \rho_L \, \frac{4}{3} \pi \overline{r^5}. \tag{59}$$



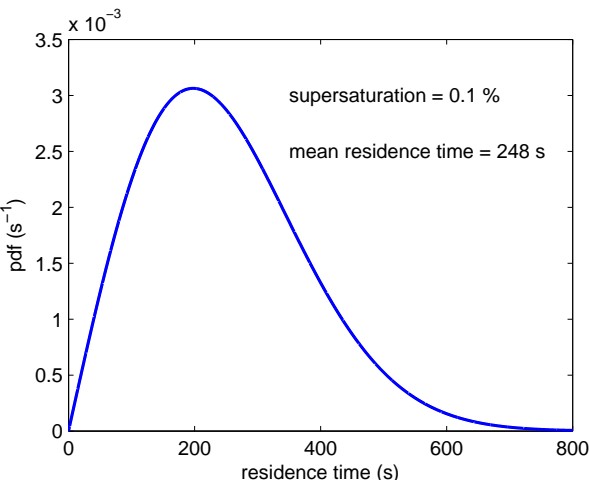

**Figure 10.** PDF of droplet residence times for 0.1% supersaturation and $h = 1$ m as given by (58).

### 7.4 Condensation rate

To derive the condensation rate of a population of droplets, $d\bar{q}/dt$, (mass of water condensed per mass of dry air per unit time), start with the condensation rate for a single droplet,

$$\frac{dm}{dt} = \rho_L 4\pi \xi r,$$

5   where we used $dr/dt = \xi/r$. Then

$$\frac{d\bar{q}}{dt} = \int_0^\infty \frac{dm}{dt} v(r) dr = D \int_0^\infty \rho_L 4\pi \xi r^2 \exp(-C r^4/4) dr = \frac{\rho_L 4\pi \xi}{\rho} N \frac{\sqrt{2}\,\Gamma(3/4)}{\sqrt{\pi}\, C^{1/4}}. \tag{60}$$

One can easily verify that $d\bar{q}/dt = P/(\rho h)$ using $P$ from (59) and $\overline{r^5}$ from (45), along with $C \equiv k_1/(\xi h)$ and $\Gamma(7/4)/\Gamma(3/4) = 3/4$. An equivalent form that is not specific to a particular DSD was derived by Korolev and Mazin (2003):

$$\frac{d\bar{q}}{dt} = \frac{N}{\rho} \int_0^\infty \frac{dm}{dt} p(r) dr = \frac{N\rho_L 4\pi \xi}{\rho} \int_0^\infty r\, p(r) dr = \frac{\rho_L 4\pi \xi}{\rho} N\bar{r}. \tag{61}$$

10   One can use (30) to show that (60) is equal to (61).

### 8   Conclusions

In a laboratory cloud chamber, such as the $\Pi$ Chamber at Michigan Technological University, it is possible to produce Rayleigh-Bénard convection by applying an unstable temperature gradient between the top and bottom water-saturated surfaces of the chamber. Supersaturation is produced by isobaric mixing within the turbulent flow. When aerosols (cloud condensation nuclei)





are injected at a constant rate, an equilibrium state is achieved in which the rate of droplet activation is balanced by the rate of droplet loss. After a droplet is activated, it continues to grow by condensation until it falls out (i.e., contacts the bottom surface).

Because supersaturation is difficult to measure when cloud droplets are present, it has not been generally possible to deter-
mine the magnitudes of the mean supersaturation and the supersaturation fluctuations in the Pi chamber under cloudy condi-
tions. Therefore, it also has not been generally possible to directly determine the relative contributions of mean and fluctuating supersaturation to the measured droplet PDFs.

We derived analytic PDFs of droplet radius and squared radius for conditions that could occur in a turbulent cloud chamber in which there is uniform supersaturation and a balance between droplet formation (activation) and loss (due to fall out). The
loss rate due to fall out is based on three assumptions: (1) The droplets are well-mixed by turbulence, in which case the $z$-coordinate of each droplet is a random variable. (2) When a droplet becomes sufficiently close to the lower boundary, the droplet's terminal velocity determines its probability of fall out per unit time. (3) A droplet's terminal velocity is proportional to its radius squared. Given the chamber height and the droplet fall speed's dependence on squared radius, the analytic PDFs are determined by the supersaturation alone.

It should be emphasized that it is only the supersaturation that directly determines the droplet radius PDF. A cloud chamber undergoing Rayleigh-Bénard convection is analogous to an ascending parcel: in both cases a forcing process continually increases the supersaturation, while droplet growth decreases it. For an ascending parcel, the forcing process is adiabatic cooling, while for a cloud chamber, it is turbulent fluxes of sensible heat and water vapor from the walls. In both cases, the time scale for condensation to decrease supersaturation is the phase relaxation time scale, which depends inversely on droplet
number concentration and mean radius (Korolev and Mazin, 2003). The quasi-steady supersaturation is determined by a balance between these two processes.

We demonstrated how the equilibrium radius distribution is realized by using a Monte Carlo method, and compared the results to some of those that were obtained analytically. A notable feature is the wide PDF of droplet residence times. This PDF determines the width of the DSD when there is uniform supersaturation: All droplets grow at the same rate, so the greater
a droplet's residence time, the larger it gets, and the more it contributes to the large-droplet tail of the PDF.

From the analytic equilibrium PDFs of radius and of squared radius, we obtained expressions for the median and mode radii. We also derived the first five moments of the radius from the analytic equilibrium PDFs, including moments for truncated DSDs (those with positive lower limits). We used statistics from a set of measured DSDs to check for consistency with the analytic PDF. We found consistency between theoretical and measured moments, but only when the truncation radius of the measured
DSDs was taken into account. Because the theoretical moments depend only on the supersaturation once the chamber height, Stokes' fall speed parameter, and truncation radius are specified, consistency between theoretical and measured moments allows us to infer the mean supersaturations that would produce the measured DSDs in the absence of supersaturation fluctuations. From the mean radius, mean squared radius, and mean cubed radius for 11 measured DSDs, the inferred mean supersaturations ranged from $0.008$ to $0.6$ %. These correspond to measured droplet number concentrations ranging from 3000 to 14 cm$^{-3}$,
respectively.





We found that accounting for the truncation radius of the measured DSDs is particularly important when comparing the theoretical and measured relative dispersions of the droplet radius. We showed that the monotonic decrease of the measured relative dispersion reported by Chandrakar et al. (2018a, c) is due to not taking truncation into account, and that when truncation of the DSD is taken into account, our theoretical values match the measured values.

5     Finally, we presented some additional quantities derived from the analytic DSD: droplet sedimentation flux, precipitation flux, and condensation rate.

*Code and data availability.*  Most of the solutions of the ordinary differential equations and integrals that appear in this study were obtained using Wolfram|Alpha (Wolfram Alpha LLC, 2019). Code used to generate the figures in this paper is available upon request. The measurements used are available in Chandrakar et al. (2018b).

10    *Author contributions.*  SKK carried out the derivations, preformed the analyses, and wrote the manuscript.

*Competing interests.*  The author declares that he has no conflicts of interest.

*Acknowledgements.*  The impetus for this study arose during a sabbatical visit to the Cloud Physics Laboratory at Michigan Technological University. The author thanks Raymond Shaw in particular for facilitating the author's visit.



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
