# Peer review of "Technical Note: Equilibrium droplet size distributions in a turbulent cloud chamber with uniform supersaturation"

_Atmospheric Chemistry and Physics, 2019_

## Referee Comment (RC1) · Anonymous Referee #2 · 18 Nov 2019

**General Comments**

The author derived analytical equilibrium solutions from the equations which govern the evolution of the droplet size distributions [Eqs. (2) and (6) in the manuscript]. The motivation of the author's study is to understand the experimental results of the socalled Pi-chamber, a laboratory cloud chamber at Michigan Technological University, which recently obtained the equilibrium droplet size distributions under the turbulent cloud conditions. To model the condition in the Pi-chamber, the author assumed (i) there is uniform supersaturation [terms with the factor  $\xi$  in Eqs. (2) and (6)], (ii) cloud droplets are activated continuously from externally injected CCNs [A(r) and B(s) in

Eqs. (2) and (6), respectively], and (iii) cloud droplets are removed from the system with the rate proportional to the droplet squared radius [terms with the factor h in Eqs. (2) and (6)]. (iii) is explained to be a simple model for the loss of cloud droplets due to sedimentation. From these assumptions, the author derived analytical solutions of cloud droplet size distributions at equilibrium state [Eqs. (12) and (17)], and also derived various analytical expressions associated with those solutions, such as various moments ( $\overline{r^n}$ , n = 1 - 5), precipitation flux, condensation rate, etc.The author then used these results to infer the condition in the Pi-chamber (Sec. 6), inferring the actual supersaturation in the Pi-chamber (from 0.008 to 0.6 % which seems to be reasonable) and also demonstrating the importance of the truncation radius of the size distributions measured in the Pi-chamber.

The form of particle loss rate which is proportional to  $-k_1r^2$  is originally proposed by the author in the present study. Based on the present study, Chandrakar et al. (2019, QJRMS, doi: 10.1002/qj.3692) has recently confirmed the validity of this form of loss rate using the experimental data. It should also be noted that the applicability of the author's analysis, such as inferring the supersaturation in the Pi-chamber and checking the importance of the truncation radius, are not necessarily limited to the case considered in the author's present study. In principle, these ideas of analysis can be applied to other cases such as under the condition of fluctuating supersaturation.

I think the author has made an original contribution and the manuscript is appropriate for the Atmospheric Chemistry and Physics. I only suggest minor revisions before acceptance as below.
**Specific Comments**

Size distribution at  $r = r_a$

From Eq. (9), it can be written as below

$$v(r_a) = \frac{r_a}{\xi} \int_{r_0}^{r_a} A(r) dr.$$

On the other hand, from the general solution Eq. (12),

$$v(r) = Dr \exp(-Cr^4/4).$$

Does this mean that Eq. (9) can be related to Eq. (12) by substituting  $r = r_a$  in Eq. (12)? I think it might be informative for readers if the author adds an explanation on how Eq. (9) is connected to the general solution Eq. (12).

Nominal supersaturation in Fig. 6

In page 14, line 4, the author cites Rogers and Yau (1989) and explains that the critical radius for injected NaCl particles is about  $r_* \sim 0.6 \mu m$ . I think the same textbook also gives an estimation of the critical supersaturation for those particles and I expect it to be about  $S_* \sim 0.1\%$ . On the other hand, according to Figure 6, the inferred nominal mean supersaturations for the Pi-chamber experiments with two largest number densities of cloud droplets are smaller than 0.01% ( $\overline{S}_{nominal}

**Technical Corrections**
1. p. 3, Sec 2.3, line 1 :  $u/h\Delta t = k_1 r^2/h\Delta t \longrightarrow (u/h)\Delta t = (k_1 r^2/h)\Delta t$ 2. p. 4, line 6 :  $k_1 r^2/h\Delta t \longrightarrow (k_1 r^2/h)\Delta t$ 3. p. 9, Figs. 3 & 4, y-axis :  $pdf(\mu m)^{-1}) \longrightarrow pdf((\mu m)^{-1}) \text{ or } pdf(\mu m^{-1})$

---

## Referee Comment (RC2) · Anonymous Referee #1 · 19 Nov 2019

**Review of "Technical Note: Equilibrium droplet size distributions in a turbulent cloud chamber with uniform supersaturation" by Krueger (acp-2019-932)**

The submitted study investigates steady-state droplet size distributions in a turbulent environment. These distributions, including several moments and other quantities, are derived analytically by considering the microphysical processes of droplet growth by diffusion and their removal by sedimentation, as well as an (artificial) droplet production term. These theoretical results are compared with recent measurements by Chandrakar et al. (2018), adding valuable information for the interpretation of the aforementioned measurements but also the quantification of droplet size distributions by measurements in general.

All in all, this well-written technical note gives new and interesting insights into the development of droplet size distributions. Overall, the manuscript is in an almost publishable state. Nonetheless, I have some very minor suggestions below. I fully support the publication of the manuscript in Atmospheric Chemistry and Physics.

**Minor Comments**

Sec. 2.4: I believe that Srivastava (1991) also requires some recognition in this subsection. He investigated, also analytically, the mean, standard deviation, and dispersion of droplet spectra, including the effects of droplet surface tension.

P. 4, ll. 28 ff.: A supersaturation of 10 % is relatively high for a typical cloud. For plotting the analytical solutions, a more realistic value of 0.1 % is used. I suggest to also use this lower supersaturation in the Mote-Carlo calculations of section 3. However, this will not change any conclusions.

P. 5, ll. 11 – 12: I would emphasize that the "stochastic nature of the droplet fallout process" includes the assumed stochastic rearrangements of the droplets along the $z$-axis, i.e., turbulent motions, since the sedimentation process itself is deterministic.

Sec. 4.2: It is possible to speed up the derivation of $w(s)$ using $v(r)$. By acknowledging that $v = dN/dr$ and $w = dN/ds = dN/dr^2$, where $N$ is the total number of droplets, we see that $dN = v\,dr = w\,dr^2$. Hence, $w = v/(dr^2/dr) = v/(2r) = D/2\exp(-Cr^4/4) = G\exp(-Cs^2/4)$, using that $G = D/2$ and $s = r^2$.

Eqs. (12) and (17): Because Eqs. (12) and (17) clearly violate the assumptions $v(r_0) = w(s_0) = 0$, it might be helpful to state – again – that the analytical solutions are only defined for $r > r_a > r_0 > 0$ and $s > s_a > s_0 > 0$.

P. 7, l. 10: The first term of the equation contains one minus sign too much.

Eq. (26): Also this deviation can be shortened: $I(R) = 1 - f(R)$ with $f(R)$ already derived in Eq. (22).

P. 21, l. 20: The usual citation for the phase relaxation timescale is Squires (1952).

**Technical Comments**

P. 2, l. 26: Throughout the paper, the author uses plural personal pronouns ("we" or "us"). Thus, this single "I" feels odd.

P. 3, l. 27; p. 4, l. 6; p. 4, l. 26: For clarity, add parentheses to the equations for the fall out probability: $(u/h)\Delta t$ and $(k_1 r^2/h)\Delta t$ instead of $u/h\Delta t$ and $k_1 r^2/h\Delta t$, respectively.

P. 4, l. 31; p. 5, l. 9: Since the analytical solution will be introduced further below, I suggest adding a "to-be-determined" in front of "analytical solution".

P. 6, l. 1: $A(r)$ has been previously introduced as the "production of (activated) droplets from the injected aerosol" (p. 3, ll. 8 – 9). Here, it is called the "production of droplets by activation". Although these processes are identical in the described framework, I suggest homogenizing the terminology.

Eq. (18): I would add a comma (",") to the end of the equation.

Figs. 3 and 4: An opening parenthesis ("(") is missing in the ordinate title.

P. 10, l. 14: I would add a comma (",") to the end of the equation.

P. 11, ll. 20 – 22: This comment feels too technical. I would omit it or state this information in a footnote.

**References**

Chandrakar, K. K., Cantrell, W., and Shaw, R. A. (2018). Influence of turbulent fluctuations on cloud droplet size dispersion and aerosol indirect effects, Journal of the Atmospheric Sciences, 75, 3191–3209.

Squires, P. (1952). The growth of cloud drops by condensation. I. General characteristics. Australian Journal of Chemistry, 5(1), 59-86

Srivastava, R. C. (1991). Growth of cloud drops by condensation: Effect of surface tension on the dispersion of drop sizes. *Journal of the Atmospheric Sciences*, *48*(13), 1596-1599.

---

## Author Response (AR1)

**Responses to reviews of "Technical Note: Equilibrium droplet size distributions in a turbulent cloud chamber with uniform supersaturation" by Krueger (acp-2019-932)**

**1 Reviewer Comments 1**

The author thanks Reviewer 1 for his/her careful reading of the manuscript and helpful suggestions and comments. All were incorporated.

**Specific Comments**

Size distribution at $r = r_a$

From Eq. (9), it can be written as below

$$v(r_a) = \frac{r_a}{\xi} \int_{r_0}^{r_a} A(r) dr.$$

On the other hand, from the general solution Eq. (12),

$$v(r) = Dr\exp(-Cr^4/4).$$

Does this mean that Eq. (9) can be related to Eq. (12) by substituting $r = r_a$ in Eq. (12)? I think it might be informative for readers if the author adds an explanation on how Eq. (9) is connected to the general solution Eq. (12).

*RESPONSE:* Eq. (12) is now solved for $D$ when $r = r_a$. The resulting expression is given by Eq. (13) which contains the boundary condition, $v(r_a)/r_a$ given by Eq. (9). The link between Eq. (12) and Eq. (9) should now be clear.

Nominal supersaturation in Fig. 6

In page 14, line 4, the author cites Rogers and Yau (1989) and explains that the critical radius for injected NaCl particles is about $r_* \sim 0.6\mu$m. I think the same textbook also gives an estimation of the critical supersaturation for those particles and I expect it to be about $S_* \sim 0.1$%. On the other hand, according to Figure 6, the inferred nominal mean supersaturations for the Pi-chamber experiments with two largest number densities of cloud droplets are smaller than $0.01$% ($\overline{S}_{\text{nominal}} < 0.01$%). This seems somehow strange, because aerosol particles cannot be activated to cloud droplets if the supersaturation $\overline{S}_{\text{nominal}}$ is much smaller than the critical supersaturation $S_*$. Does the author have possible explanations for this apparent discrepancy? If so, providing those explanations in the manuscript might be helpful for readers.

*RESPONSE:* This comment motivated an important extension to the original study. The apparent discrepancy occurs because the analytic solution omits droplet curvature and solute effects, and therefore exhibits no dependence on aerosol properties, and does not have a critical supersaturation. In response, I modified the manuscript extensively in section 6.

On page 15, I gave a more complete characterization of the aerosol size distributions used in the cloud chamber and stated that "The potential impacts of both droplet curvature and solute effects on comparisons of analytic and measured DSDs will be discussed below, in section 6.2."

I added section 6.2 (pages 17–19) on inferred mean supersaturation and droplet activation. In this section, I noted that 99% of the injected aerosol particles have a dry diameter less than about 170 nm. The critical supersaturation for a NaCl particle with a dry diameter of 170 nm is 0.052%. I also noted that 6 DSDs in Figure 6 have inferred supersaturations less than 0.052%. The rest of section 6.2 discusses the implications of this situation. I discussed the following possibilities (excerpted from the revised manuscript):

1. Neglecting droplet curvature and solute effects in the analytic DSD governing equation produces significant underestimates of the inferred supersaturations. It could be that once curvature and solute effects are included in the droplet growth equation, the inferred mean supersaturations for all 11 measured DSDs will be large enough to activate at least the largest of the injected aerosols.

   To investigate this possibility, we used the droplet growth equation, both with and without the curvature and solute terms included, in the Monte Carlo model described in section 3 to calculate mean droplet radius versus supersaturation for 100 supersaturation values (Figure 8). Figure 8 shows that the mean droplet radius is smaller when these terms are included, for the same fixed supersaturation. This is due to the slower initial growth of the droplets. The differences in mean radius are largest for supersaturations slightly larger than the critical supersaturation.

How do the curvature and solute terms affect the inferred supersaturation? For a given droplet radius, the inferred supersaturation is larger with solute and curvature terms included. In our specific case, Figure 8 suggests that a measured DSD ($r > 2.5$ $\mu$m only) with a mean radius of about 4.4 $\mu$m or larger could have been activated and grown with a fixed supersaturation of 0.055%. Figure 6 shows that this requirement excludes the measured DSDs with the 5 smallest mean radii.

2. Even after including droplet curvature and solute effects, the inferred supersaturations of the 5 measured DSDs with the smallest mean radii are less than the critical supersaturation of the largest of the injected aerosols. In this case, we conclude that there must have been supersaturation fluctuations somewhere in the cloud chamber that exceeded the critical supersaturation for at least the larger injected aerosols. There are two possible situations:

   (a) Large supersaturation fluctuations occur only near the bottom and top boundaries of the cloud chamber, as is typical of Rayleigh-Bénard convection. In this case, it could be that activated droplets are transported away from the boundaries and then continue to grow consistent with inferred mean supersaturations calculated with droplet curvature and solute effects included. This scenario is analogous to droplets growing in a cumulus updraft: The droplets are activated by relatively large supersaturations just above cloud base, but then continue to grow in lower supersaturations at higher levels (Rogers and Yau 1989).

   (b) Droplet growth in the chamber for these DSDs is primarily or entirely due to supersaturation fluctuations throughout the cloud chamber. In this case, the analytic DSD solution, which assumes that there are no supersaturation fluctuations, is not valid. Chandrakar et al. (2020b) found that analytic solutions for DSDs when mean supersaturation is absent (but fluctuations are present) have nearly the same shape as DSDs for no supersaturation fluctuations. As a result, it is difficult to distinguish the two cases based only on the consistency of the moments.

I also revised the conclusions by adding the following text to pages 25–26:

We found that neglecting the curvature and solute terms in the droplet growth rate equation can sometimes affect the inferred supersaturations. For a given droplet radius, the inferred supersaturation is larger with solute and curvature terms included. Calculations with a Monte Carlo model with solute and curvature terms included suggest that for the aerosols injected into the cloud chamber, a measured DSD ($r > 2.5$ $\mu$m only) with a mean radius of about 4.4 $\mu$m or larger could have been activated and grown with a fixed supersaturation of 0.055%. This excludes the DSDs with the 5 smallest mean radii. To produce these DSDs, there must have been supersaturation fluctuations somewhere in

the cloud chamber that exceeded the critical supersaturation for at least the larger injected aerosols.

**Technical Corrections**

1. p. 3, Sec 2.3, line 1     :    $u/h\Delta t = k_1 r^2/h\Delta t$    $\longrightarrow$    $(u/h)\Delta t = (k_1 r^2/h)\Delta t$
2. p. 4, line 6               :        $k_1 r^2/h\Delta t$       $\longrightarrow$         $(k_1 r^2/h)\Delta t$
3. p. 9, Figs. 3 & 4, y-axis   :     $\mathsf{pdf}(\mu\mathsf{m})^{-1})$      $\longrightarrow$    $\mathsf{pdf}((\mu\mathsf{m})^{-1})$ or $\mathsf{pdf}(\mu\mathsf{m}^{-1})$

*All done.*

**2 Reviewer Comments 2**

The author thanks Reviewer 2 for his/her careful reading of the manuscript and helpful suggestions and comments. Essentially all were incorporated.

**Minor Comments**

**Sec. 2.4** I believe that Srivastava (1991) also requires some recognition in this subsection. He investigated, also analytically, the mean, standard deviation, and dispersion of droplet spectra, including the effects of droplet surface tension.

*Srivastava (1991) is now cited in section 2.1 (p. 3, line 9).*

**P. 4, ll. 28 ff.** A supersaturation of 10 % is relatively high for a typical cloud. For plotting the analytical solutions, a more realistic value of 0.1 % is used. I suggest to also use this lower supersaturation in the Monte-Carlo calculations of section 3. However, this will not change any conclusions.

*Done. (Figures 1 and 2 replaced.)*

**P. 5, ll. 11–12** I would emphasize that the "stochastic nature of the droplet fallout process" includes the assumed stochastic rearrangements of the droplets along the z-axis, i.e., turbulent motions, since the sedimentation process itself is deterministic.

*Done. (P. 5, last sentence.)*

**Sec. 4.2** It is possible to speed up the derivation of $w(s)$ using $v(r)$...

*Replaced original derivation with this one (Sec. 4.2, pp. 7–8).*

**Eqs. (12) and (17)** Because Eqs. (12) and (17) clearly violate the assumptions $v(r_0) = w(s_0) = 0$, it might be helpful to state—again—that the analytical solutions are only defined for $r > r_a > r_0 > 0$ and $s > s_a > s_0 > 0$.

*Done. (P. 8. line 12).*

**P. 7, l. 10** The first term of the equation contains one minus sign too much.

*Eq. no longer included because of comment on Sec 4.2.*

**Eq. (26)** Also this deviation can be shortened: . . .

*Done. (Eq. (22) on p. 10.)*

**P. 21, l. 20** The usual citation for the phase relaxation timescale is Squires (1952).

*Replaced. (P. 25, line 10.)*

**Technical Comments**

P. 2, l. 26: Throughout the paper, the author uses plural personal pronouns ("we" or "us"). Thus, this single "I" feels odd.

P. 3, l. 27; p. 4, l. 6; p. 4, l. 26: For clarity, add parentheses to the equations for the fall out probability: $(u/h)\Delta t$ and $(k_1 r^2/h)\Delta t$ instead of $u/h\Delta t$ and $k_1 r^2/h\Delta t$, respectively.

P. 4, l. 31; p. 5, l. 9: Since the analytical solution will be introduced further below, I suggest adding a "to-be-determined" in front of "analytical solution".

P. 6, l. 1: $A(r)$ has been previously introduced as the "production of (activated) droplets from the injected aerosol" (p. 3, ll. $8-9$). Here, it is called the "production of droplets by activation". Although these processes are identical in the described framework, I suggest homogenizing the terminology.

Eq. (18): I would add a comma (",") to the end of the equation.

Figs. 3 and 4: An opening parenthesis ("(") is missing in the ordinate title.

P. 10, l. 14: I would add a comma (",") to the end of the equation.

P. 11, ll. $20-22$: This comment feels too technical. I would omit it or state this information in a footnote.

*All of these were done except for the last one. The ACP Author's Guide states that footnotes should be avoided, as they tend to disrupt the flow of the text. In addition, there was not a good place to add a footnote.*

**3   Notes on the marked-up manuscript version**

Following is a marked-up manuscript version showing the changes made (using latexdiff in LaTeX).

Blue indicates added text, red indicates deleted text. New references are highlighted in yellow.

Please note that:

1. Figures 1 and 2 are new versions of the originals, as requested by Reviewer 1, even though this is not evident in the marked-up manuscript.

2. The location of Figure 6 in the LaTeX file was moved, but the caption was NOT revised despite the blue color.

3. The location of Figure 7 was also moved, but the caption was not changed, despite the red and blue colors.

4. Figure 8 is new, as is its entire caption.

5. Section 8.1 Liquid water content is new (added on my own volition)

6. Section 8.3 Precipitation flux was moved and now immediately follows Droplet sedimentation flux.

[revised manuscript text omitted]

$$\text{w} = \frac{\text{v}}{dr^2/dr} = \frac{\text{v}}{2r} = \frac{\text{D}}{2}\exp(-\text{C}\,r^4/4) = \text{G}\exp(-\text{C}\,s^2/4) \tag{14}$$

using (12), $s = r^2$, and $G = D/2$. Just as for the o.d.e. instead of (??) for $s_a < s < \infty$:

$$0 = -2\xi\frac{dw}{ds} - w\frac{k_1}{h}s,$$

with the boundary condition at $s = s_a$ given by (??) . When the supersaturation is steady, $\xi$ is a constant so we can write (??) as

$$0 = -\frac{dw}{ds} - \frac{C}{2}ws.$$

The general solution to (??) is

$$w(s) = G\exp(-Cs^2/4)$$

where $G$ is an integration constant with units of $(\text{mass})^{-1}(\text{length})^{-2}$. (10), the corresponding solution ( gen-sol) is valid only for $r > r_a > r_0 > 0$. Similarly, (14) is valid only for $s > s_a > s_0 > 0$.

**4.3   Droplet number concentration and integration constantsconstant**

As already noted, $v(r)\,dr$ $v(r)\,dr$ is the number of cloud droplets per unit mass of air with radii in the interval $(r, r+dr)$ $[r, r+dr]$. Therefore, the number of cloud droplets per unit *volume* of air is

$$N = \rho\int_0^\infty v(r)\,dr = \rho D\int_0^\infty r\exp(-Cr^4/4)\,dr = \rho D\frac{\sqrt{\pi}}{2\sqrt{C}}$$

$$\text{N} = \rho\int_0^\infty \text{v(r)}\,\text{dr} = \rho\text{D}\int_0^\infty \text{r}\exp(-\text{C}\,r^4/4)\,\text{dr} = \rho\text{D}\frac{\sqrt{\pi}}{2\sqrt{\text{
[revised manuscript text omitted]